# Association between Mean Arterial Pressure during the First 24 Hours and Clinical Outcome in Critically Ill Stroke Patients: An Analysis of the MIMIC-III Database

**DOI:** 10.3390/jcm12041556

**Published:** 2023-02-16

**Authors:** Sheng Zhang, Yun-Liang Cui, Sheng Yu, Wei-Feng Shang, Jie Li, Xiao-Jun Pan, Zhen-Liang Wen, Si-Si Huang, Li-Min Chen, Xuan Shen, Yue-Tian Yu, Jiao Liu, De-Chang Chen

**Affiliations:** 1Department of Critical Care Medicine, Ruijin Hospital, Shanghai Jiao Tong University School of Medicine, Shanghai 200025, China; 2Department of Critical Care Medicine, The 960th Hospital of the PLA Joint Logistics Support Force, Jinan 250000, China; 3Department of Critical Care Medicine, Changshu Second People’s Hospital, Changshu 215500, China; 4Department of Laparoscope Surgery, 986 Hospital of People’s Liberation Army Air Force, Xi’an 719000, China; 5Department of Critical Care Medicine, Renji Hospital, Shanghai Jiao Tong University School of Medicine, Shanghai 200025, China

**Keywords:** mean arterial pressure, stroke, mortality, risk factor, intensive care unit, restricted cubic splines

## Abstract

Abnormal blood pressure is common in critically ill stroke patients. However, the association between mean arterial pressure (MAP) and mortality of critically ill stroke patients remains unclear. We extracted eligible acute stroke patients from the MIMIC-III database. The patients were divided into three groups: a low MAP group (MAP ≤ 70 mmHg), a normal MAP group (70 mmHg < MAP ≤ 90 mmHg), and a high MAP group (MAP > 90 mmHg). The Cox proportional hazards model and restricted cubic splines were used to assess the association between MAP and mortality. Sensitivity analyses were conducted to investigate whether MAP had different effects on mortality in different subpopulations. A total of 2885 stroke patients were included in this study. The crude 7-day and 28-day mortality was significantly higher in the low MAP group than that in the normal MAP group. By contrast, patients in the high MAP group did not have higher crude 7-day and 28-day mortality than those in the normal MAP group. After multiple adjustments using the Cox regression model, patients with low MAP were consistently associated with higher 7-day and 28-day mortality than those with normal MAP in the following subgroups: age > 60 years, male, those with or without hypertension, those without diabetes, and those without CHD (*p* < 0.05), but patients with high MAP were not necessarily associated with higher 7-day and 28-day mortality after adjustments (most *p* > 0.05). Using the restricted cubic splines, an approximately L-shaped relationship was established between MAP and the 7-day and 28-day mortality in acute stroke patients. The findings were robust to multiple sensitivity analyses in stroke patients. In critically ill stroke patients, a low MAP significantly increased the 7-day and 28-day mortality, while a high MAP did not, suggesting that a low MAP is more harmful than a high MAP in critically ill stroke patients.

## 1. Introduction

Stroke is a leading cause of death and disability worldwide and can be broadly classified into ischemic stroke and hemorrhagic stroke, the latter of which includes intracerebral hemorrhage and subarachnoid hemorrhage [1,2]. Each year, a considerable number of critically ill stroke patients are admitted to intensive care units (ICUs), resulting in a growing burden of health-care globally.

Hypertension has been proved to be one of the most important risk factors that contribute to the occurrence of stroke [3,4], and the majority of patients often have elevated BP in the acute phase of stroke. Currently, researchers propose that BP elevation is a response to brain ischemia, brain edema, increased sympathetic activation, and/or mental stress [1,3,4]. However, the relationship between BP after stroke onset and prognosis has not been fully established [5,6]. Several observational studies found that elevated BP was associated with poor prognosis after stroke, while other studies have revealed a more complex relationship between the two [7,8,9]. In addition, confounding factors, including age, sex, the scale of hematoma, and the level of consciousness, were rarely taken into account [10]. Although it is generally accepted that BP management is important for stroke patients, the strategies and approaches for BP management vary depending on stroke type and time after stroke onset. There are still questions to be investigated, such as the optimal timing to initiate BP control and the targeted BP level in disease-specific populations [2].

In the present study, we sought to explore the association between MAP and the 7-day and 28-day mortality in critically ill patients with stroke, and to determine whether this association varies among disease-specific populations.

## 2. Methods

### 2.1. Data Sources

In this study, all data were extracted from the MIMIC-III database. The MIMIC-III database is a public intensive-care database that contains comprehensive, time-stamped information for more than 50,000 ICU admissions at Beth Israel Deaconess Medical Center (BIDMC) in Boston, Massachusetts between 2001 and 2012, representing more than 38,000 unique patients. In this database, all the data of patients were structured and stored in 26 tables based on demographic characteristics, vital signs, laboratory and imaging results, and outcomes. We were granted access to the MIMIC-III database after completion of the course and test of data use agreement (certification number: 42442549). As the MIMIC-III database has received ethical approval from the Institutional Review Boards (IRBs) at BIDMC and MIT, an extra ethical consent was waived, as previously described [11].

### 2.2. Study Population

We screened 53,423 distinct hospital admissions in the MIMIC-III database and included patients if the following criteria were met: (1) age ≥ 18 years on ICU admission; and (2) the first diagnosis ICD-9 code ranged from 430 to 434, which indicated that the primary diagnosis was stroke (including subarachnoid hemorrhage, intracerebral hemorrhage, other and unspecified intracranial hemorrhage, occlusion and stenosis of precerebral arteries, occlusion of cerebral arteries) [12]. The exclusion criteria were: (1) if the data on clinical outcomes and the length of hospital stay were missing; and (2) patients who died or were discharged within 24 h after ICU admission. For patients who were admitted to ICU more than once, only the first scenario was analyzed.

### 2.3. Data Extraction and Variables

We used Navicat Premium version 12.1.28 (PremiumSoft CyberTech Ltd., Hong Kong, China) and SQL (Structured Query Language) to extract data from the MIMIC-III database. The included variables were: demographic features (age, gender, weight, and ethnicity), comorbidities (hypertension, cardiac arrhythmias, chronic heart disease, peptic ulcer, liver disease, diabetes, chronic obstructive pulmonary disease, coagulopathy, solid tumor, and the Elixhauser Comorbidity Index), vital signs within 24 h after ICU admission (heart rate, respiratory rate, temperature, and mean arterial pressure), interventions (mechanical ventilation and renal replacement therapy), laboratory results [PaO2, PaCO2, pH, white blood cell count (WBC), platelet, hemoglobin, glucose, creatinine, chloride, sodium, potassium, blood urea nitrogen (BUN), international normalized ratio (INR), prothrombin time (PT), activated partial thromboplastin time (APTT), and bicarbonate], and severity of organ dysfunction [the Simplified Acute Physiology Score (SAPS) and the Sequential Organ Failure Assessment (SOFA)].

### 2.4. Endpoints and Definitions

The primary endpoint was defined as the 7-day and 28-day mortality from the date of ICU admission. MAP was defined as the mean arterial pressure during the first 24 h after admission. The low MAP group, the normal MAP group, and the high MAP group were defined as MAP value ≤ 70 mmHg, 70 mmHg < MAP ≤ 90 mmHg, and MAP > 90 mmHg, respectively. The definition of comorbidities, including hypertension, diabetes, solid tumor, coagulopathy, and obesity, were according to the criteria proposed by Elixhauser and colleagues [13]. Renal replacement therapy (RRT) was defined as any form of extracorporeal renal support or replacement therapy.

### 2.5. Statistical Analysis

Continuous variables were presented as mean and standard deviation (SD) or median (interquartile range) and compared by ANOVA followed by Tukey’s multiple comparisons test (normal-distributed) or the Kruskal-Wallis test and Benjamini & Hochberg method (non-normal-distributed). Categorical variables were presented as numbers (percentages) and compared using a chi-square test or Fisher exact test, as appropriate. To analyze the relationship between MAP and mortality in stroke patients, we first divided all the patients into three groups according to the MAP values as abovementioned. After that, we applied five Cox proportional hazards models to determine whether a low or a high MAP was associated with an increased risk of 7-day and 28-day mortality after adjusting for different sets of confounders. Model one adjusted for age, sex, and ethnicity. Model two adjusted for model one plus heart rate, temperature, PCO2, and pH. Model three adjusted for model two plus glucose, platelet, creatinine, and BUN. Model four adjusted for all covariates except SOFA score and SAPS score. Model five adjusted for all covariates. The results are presented as hazard ratios (HRs) and 95% confidence intervals (CIs). Given that the association between MAP and mortality could be nonlinear, we performed restricted cubic spline analyses. To achieve this, MAP was used as a continuous variable and fitted into the Cox proportional hazards model using the cubic spline function to account for a possible nonlinear relationship between MAP and outcomes. Finally, we performed sensitivity analyses to test the robustness of our primary findings. To this aim, we repeated multivariable Cox proportional hazards models and restricted cubic spline analyses to determine whether the impact of MAP on mortality was consistent across different subgroups, including age > 60 or ≤60 years, male or female, with or without hypertension, with or without diabetes, and with or without CHD. To avoid overfitting and to enhance internal validation, we used 5-fold cross-validation. We processed and analyzed all the data by R software (Version 3.6.2). A two-tailed *p* < 0.05 was considered statistically significant.

## 3. Results

### 3.1. Baseline Characteristics

Table 1 shows the baseline characteristics of patients according to MAP stratifications. A total of 2885 stroke patients were included in this study. Among them, 289, 1794, and 802 patients were in the low MAP group, the normal MAP group, and the high MAP group, respectively.

Compared with the normal MAP group, patients in the low MAP group had significantly higher levels of SAPS scores, creatine, BUN, potassium, APTT, and PT (all *p* < 0.05), and had significantly lower levels of heart rate, temperature, hemoglobin, and weight (all *p* < 0.05). In contrast, patients in the high MAP group had significantly higher levels of heart rate, respiratory rate, hemoglobin, potassium, and weight (all *p* < 0.05), and had significantly lower levels of glucose, SOFA scores, and SAPS scores (all *p* < 0.05) than those in the normal MAP group. Compared with the normal MAP group, patients in the low and high MAP groups did not exhibit a significant difference among variables of PaO2, platelet, WBC, chloride, bicarbonate, Elixhauser scores, RRT dependence, requirement of mechanical ventilation, the proportion of liver disease, solid tumor, and coagulopathy.

The crude 7-day and 28-day mortality was significantly higher in the low MAP group than that in the normal MAP group [7-day mortality (25.6% vs. 15.5%, *p* < 0.001); 28-day mortality (29.8% vs. 20.4%, *p* < 0.001)]. By contrast, patients in the high MAP group did not have higher crude 7-day and 28-day mortality than those in the normal MAP group [7-day mortality (17.2% vs. 15.5%, *p* = 0.298); 28-day mortality (22.4% vs. 20.4%, *p* = 0.259)].

### 3.2. Relationship between MAP and Mortality

As shown in the Kaplan-Meier plot (Figure 1), the cumulative 7-day and 28-day survival probabilities were significant lower in the low MAP group than those in the normal MAP group (all *p* < 0.001), but the cumulative survival probabilities did not significantly differ between the normal MAP group and the high MAP group (28-day *p* = 0.272; 7-day *p* = 0.311).

After adjustment for age, sex, and ethnicity using the multivariable Cox proportional hazards model (Model one), a low MAP was associated with higher 7-day [hazard ratio (HR), 1.71; 95% CI, 1.32–2.12; *p* < 0.001] and 28-day (HR, 1.51; 95% CI, 1.19–1.91; *p* < 0.001) mortality when compared with the normal MAP group. When further adjustment was conducted using the covariates listed from model two to model five, respectively, the association between a low MAP and 7-day and 28-day mortality remained significant (Table 2). By contrast, a high MAP was not associated with higher 7-day and 28-day mortality when compared with the normal MAP in most models (from model one to four), with the exception that in model five, a high MAP was associated with increased risk of 7-day (HR, 1.26; 95% CI, 1.01–1.57; *p* = 0.043) and 28-day (HR, 1.28; 95% CI, 1.06–1.55; *p* = 0.012) mortality.

### 3.3. Non-Linear Association between MAP and Outcomes

Restricted cubic spline analyses were used to visualize the association between MAP and mortality. An approximately L-shaped non-linear association between MAP and mortality (with 80 mmHg as a reference) was observed (Figure 2). The risk of mortality decreased rapidly before MAP reached 80 mmHg and then started to become relatively flat afterwards (*p* for non-linearity test < 0.001). The results were consistent after adjustment for all the covariates listed in model five (Figure 3), suggesting that the harmful effects caused by low MAP were more pronounced than those caused by high MAP in stroke patients.

### 3.4. Sensitivity Analysis

To further verify the association between MAP and mortality in stroke patients, sensitivity analyses using the multivariable Cox regression models were performed in subgroups with different age, gender, and with or without comorbidities (hypertension, diabetes, and CHD). The association between alow MAP and the 7-day and 28-day mortality was consistent in the following subgroups: age > 60 years, male, those with or without hypertension, those without diabetes, and those without CHD (Figure 4). Then we re-performed restricted cubic spline analyses in the same above-mentioned subgroups; the approximately L-shaped association between MAP and the 7-day and 28-day mortality remained robust across different subgroups (Figure 5 and Figure 6), indicating that a low MAP was more strongly associated with mortality compared with a high MAP in varied sub-populations.

## 4. Discussion

The present study aimed to investigate the relationship between MAP and mortality in patients with acute stroke. Unlike other clinical studies, all data in our study were extracted from a real-world database. Our main finding was that a low MAP was strongly associated with an increased risk of 7-day and 28-day mortality, whereas a high MAP had less significant impact on mortality.

Stroke unit care is a key component of the World Stroke Organization’s global guidelines and is likely the intervention with the greatest overall benefits that can reduce both morbidity and mortality for all stroke patients [14]. Despite the substantial advances in therapy that have occurred in the past 5 years [15], many questions remain unresolved, including the optimal blood pressure target [16,17]. Population-based studies have shown that 75% or more of stroke patients have elevated blood pressure and 5% have decreased or normal BP [2,8,18]. Different studies have led to different conclusions about the relationship between BP and outcomes in stroke patients [6,7,19]. Most of the previous studies and meta-analyses found that high BP in acute stroke was associated with poor outcomes [5,10,20], while some studies found that reduction in blood pressure during the acute phase provided no benefit with respect to short- and long-term mortality [21,22,23]. Currently, opinions on the effects of blood pressure lowering in acute stroke patients are controversial, and the optimal targeted BP remains unclear [24,25,26]. There are different guidelines for ischemic and hemorrhagic stroke in terms of BP management. For in-hospital patients with acute ischemic stroke, hypertension should be treated early when comorbid conditions require urgent antihypertensive treatment (e.g., concomitant acute coronary event, acute heart failure, aortic dissection, postfibrinolysis sICH, or preeclampsia/eclampsia) [27]. For patients with BP ≥ 220/120 mm Hg who did not receive IV alteplase or mechanical thrombectomy and did not have comorbid conditions that require urgent antihypertensive treatment, it is uncertain whether it is beneficial to initiate or reinitiate antihypertensive treatment within the first 48 to 72 h. It may be reasonable to reduce blood pressure by 15% within 24 h after stroke [17]. For patients with BP < 220/120 mm Hg who did not receive IV alteplase or mechanical thrombectomy and did not have comorbid conditions that require urgent antihypertensive treatment, BP lowering therapy within the first 48 to 72 h after acute ischemic stroke initiation is not associated with improved mortality or functional outcomes [17]. For patients with spontaneous ICH requiring acute BP lowering, careful titration to ensure continuous stable and sustained control of BP, avoiding SBP peaks and large variability, is beneficial for improving functional outcomes. In patients with spontaneous ICH who require BP lowering therapy, it is recommended to initiate treatment 2 h after spontaneous ICH onset and to reach the target BP within 1 h, which is associated with reduced risk of hematoma expansion and improved functional outcome [27]. In patients with mild to moderate severity of spontaneous ICH and with SBP between 150 and 220 mmHg, it is recommend to reduce the SBP to 140 mmHg rapidly and to maintain the SBP in the range between 130 and 150 mmHg, whereas acute lowering of SBP to <130 mmHg is potentially harmful and should be avoided [27]. In patients with spontaneous ICH who have large or severe ICH or require depressive surgery, the safety and efficacy of receiving intensive BP lowering is still unclear [27]. Recent studies do not support routinely lowering BP intensively in patients with acute hemorrhagic and ischemic stroke [28], because lowering blood pressure to 120 mmHg was not beneficial and led to increased renal adverse events [22,28]. Using the restricted cubic splines analyses in our study, we found a similar phenomenon: that a low MAP (<70 mmHg) significantly increased the risk of 7-day and 28-day mortality, while a high MAP (>90 mmHg) did not. We also depicted a novel non-linear relationship between MAP and the short-term mortality, with the optimal targeted MAP of 80 mmHg, where the hazard ratio for death reached the minimum value.

Previous studies have found that the prognostic significance of low BP was modified by age, gender, and comorbidities (including diabetes, hypertension, and CHD) [9,29]. Therefore, we conducted sensitivity analyses to account for potential interactions between MAP and the above-mentioned covariates. The results consistently showed an L-shaped association between MAP and mortality across different subgroups, suggesting that low MAP should be avoided in given subgroups.

There were several limitations in our study. First, although multiple analytic strategies were performed to minimize the impact of confounders on mortality, we could not account for all the confounders due to the retrospective nature of the study. For example, some variables (including surgery, medication, and imaging results) were not easy to extract, and thus were not included in the models. However, this limitation was partially addressed through sensitivity analyses. Second, as the diagnosis of stroke was identified by ICD-9 instead of clinical diagnostic criteria, a small proportion of patients may be lost during the process of data extraction. Third, without imaging results in the database, we could not further explore whether the results were consistent between patients with ischemic stroke and those with hemorrhagic stroke.

## 5. Conclusions

In critically ill stroke patients, a low MAP (<70 mmHg) significantly increased the risk of 7-day and 28-day mortality, while a high MAP (>90 mmHg) did not significantly affect the mortality. There is an approximately L-shaped association between MAP and mortality in critically ill patients with stroke.

## Figures and Tables

**Figure 1 jcm-12-01556-f001:**
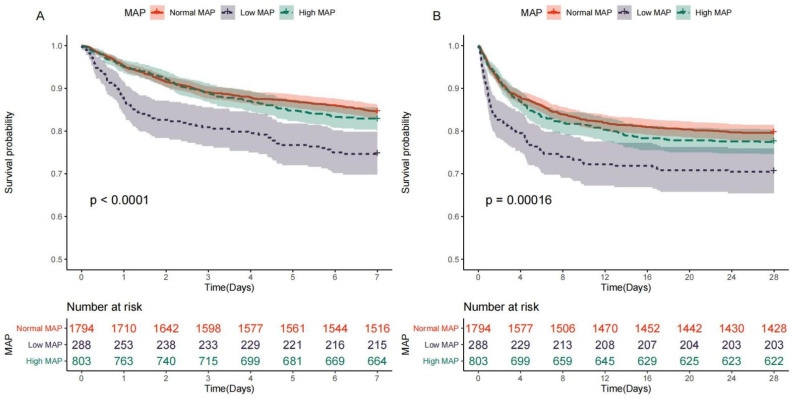
Kaplan-Meier (KM) survival curves for stroke patients with different ranges of MAP. (**A**) Kaplan–Meier survival curves for normal, high, and low MAP at day 7. (**B**) Kaplan–Meier survival curves for normal, high, and low MAP at day 28.

**Figure 2 jcm-12-01556-f002:**
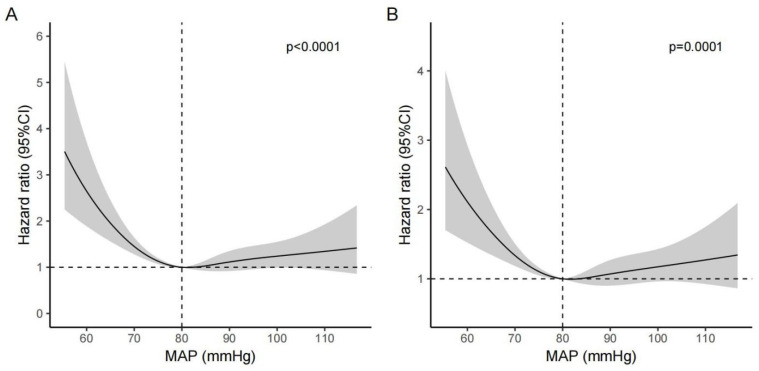
Crude non-linear association between MAP and the 7-day mortality (**A**) and 28-day mortality (**B**) in patients with acute stroke. The reference (hazard ratio = 1, horizontal dotted line) was an MAP of 80 mmHg (vertical dotted line).

**Figure 3 jcm-12-01556-f003:**
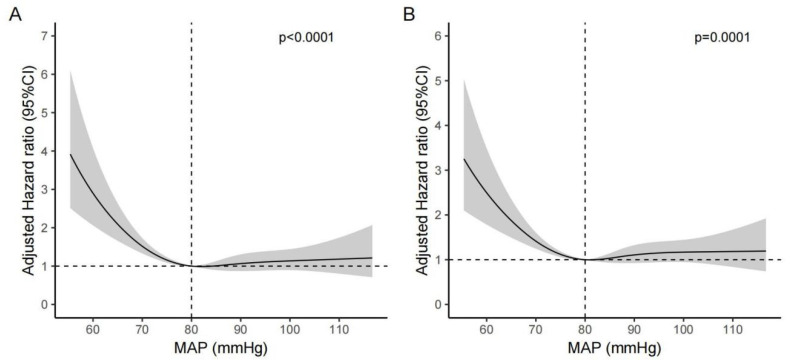
Adjusted non-linear association between MAP and the 7-day mortality (**A**) and 28-day mortality (**B**) in patients with acute stroke. All the covariates were included for adjustment.

**Figure 4 jcm-12-01556-f004:**
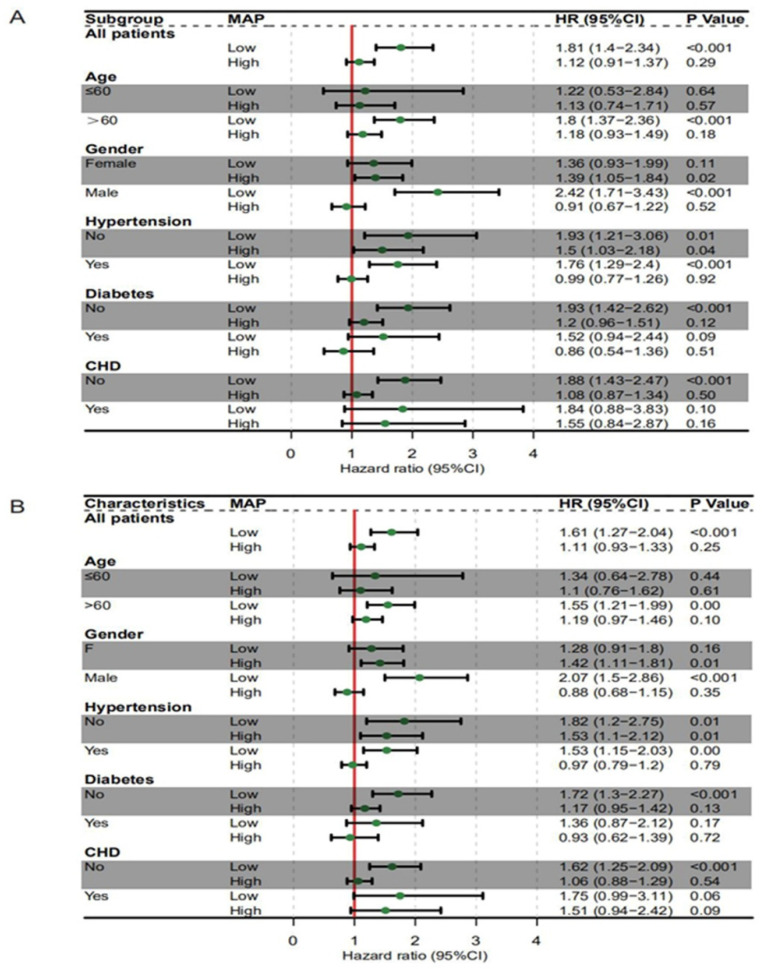
Subgroups analysis of the 7-day mortality (**A**) and 28-day mortality (**B**) according to different MAP ranges. The multivariable Cox regression model adjusted for all the covariates was performed to calculate the hazard ratio and 95% CI.

**Figure 5 jcm-12-01556-f005:**
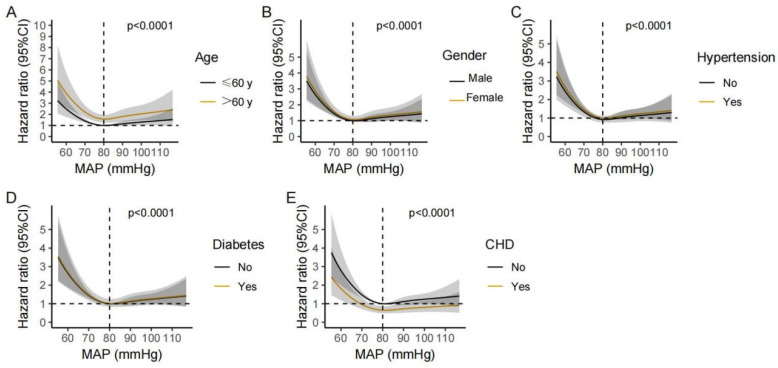
Non-linear association between MAP and the 7-day mortality of stroke patients in different subgroups. Hazard ratio and 95% CI were visualized using ribbons for following subgroups: age > 60 or ≤60 (**A**), male or female (**B**), with or without hypertension (**C**), with or without diabetes (**D**), and with or without CHD (**E**).

**Figure 6 jcm-12-01556-f006:**
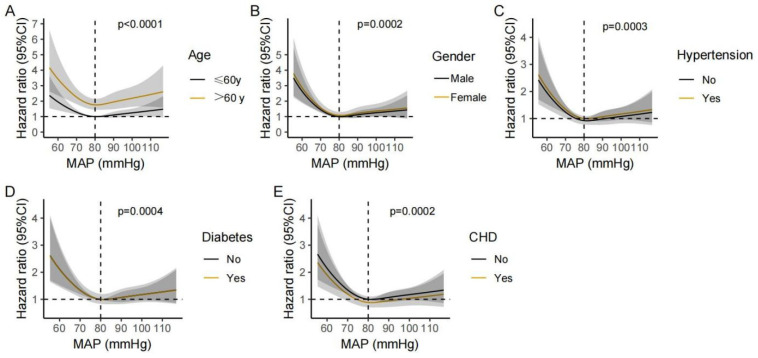
Non-linear association between MAP and the 28-day mortality of stroke patients in different subgroups. Hazard ratio and 95% CI were visualized using ribbons for following subgroups: age > 60 or ≤60 (**A**), male or female (**B**), with or without hypertension (**C**), with or without diabetes (**D**), and with or without CHD (**E**).

**Table 1 jcm-12-01556-t001:** Characteristics of acute stroke patients with normal MAP, low MAP, and high MAP on admission.

Characteristics	Overall	Normal MAP Group (70 mmHg > MAP ≤ 90 mmHg)	Low MAP Group (MAP ≤ 70 mmHg)	High MAP Group (MAP > 90 mmHg)	*p* Value for Overall	*p* Value for Normal MAP VS. Low MAP	*p* Value for Normal MAP VS. High MAP
	*n* = 2885	*n* = 1794	*n* = 289	*n* = 802			
Male	1490 (51.6%)	914 (50.9%)	128 (44.3%)	448 (55.9%)	0.002	0.042	0.035
Age, years	69.9 (57.7;79.6)	70.9 (58.1;79.7)	75.4 (66.5;81.3)	65.6 (54.7;77.6)	<0.001	<0.001	<0.001
White	2105 (73.0%)	1340 (74.7%)	230 (79.6%)	535 (66.7%)	0.001	0.493	0.003
GCS	14.0 (12.0;15.0)	14.0 (12.0;15.0)	15.0 (12.0;15.0)	14.0 (11.0;15.0)	0.023	0.376	0.039
PO2, mmHg	158 (107;237)	157 (108;238)	166 (103;257)	158 (108;227)	0.723	0.689	0.689
PCO2, mmHg	38.0 (34.0;43.0)	38.0 (33.2;43.0)	38.0 (34.0;45.0)	38.0 (34.0;42.0)	0.042	0.028	0.693
pH	7.41 (7.36;7.45)	7.41 (7.36;7.45)	7.40 (7.35;7.45)	7.41 (7.38;7.45)	<0.001	0.072	0.002
Heart rate, bpm	77.0 (68.0;87.0)	76.7 (68.0;86.5)	71.6 (64.1;82.0)	79.1 (70.6;90.1)	<0.001	<0.001	<0.001
MAP, mmHg	83.7 (76.3;91.1)	81.3 (76.5;85.5)	66.5 (63.4;68.3)	96.3 (92.8;101)	0	<0.001	0
Respiratory rate, bpm	17.7 (15.9;19.9)	17.5 (15.8;19.6)	17.8 (16.0;20.0)	18.0 (16.1;20.2)	0.001	0.286	<0.001
Temperature, °C	36.9 (36.5;37.3)	36.9 (36.5;37.3)	36.8 (36.4;37.2)	36.9 (36.5;37.3)	0.024	0.027	0.387
Glucose, mg/dL	133 (115;160)	135 (116;162)	131 (111;160)	130 (113;157)	0.003	0.108	0.004
Hemoglobin, g/L	11.7 (10.3;13.0)	11.6 (10.3;12.9)	10.6 (8.90;12.0)	12.4 (11.0;13.6)	<0.001	<0.001	<0.001
Platelet, ×10^9^/L	214 (170;266)	212 (170;265)	213 (157;264)	220 (176;270)	0.054	0.337	0.067
WBC, ×10^9^/L	11.4 (8.70;14.7)	11.4 (8.70;14.8)	11.2 (8.40;15.0)	11.5 (8.90;14.3)	0.918	0.898	0.898
Creatinine, mg/dL	0.90 (0.80;1.20)	0.90 (0.80;1.20)	1.00 (0.80;1.40)	1.00 (0.80;1.20)	0.002	0.001	0.27
BUN, mg/dL	18.0 (13.0;25.0)	18.0 (13.0;24.0)	20.0 (15.0;32.0)	17.0 (13.0;23.0)	<0.001	<0.001	0.526
Potassium, mmol/L	3.70 (3.40;4.00)	3.70 (3.40;4.00)	3.80 (3.50;4.10)	3.60 (3.30;4.00)	<0.001	0.003	0.005
Sodium, mmol/L	138 (136;140)	138 (136;140)	138 (135;140)	138 (136;140)	0.005	0.565	0.007
Chloride, mmol/L	103 (100;106)	103 (100;106)	103 (100;106)	103 (100;105)	0.778	0.72	0.72
Bicarbonate, mmol/L	23.0 (21.0;26.0)	23.0 (21.0;26.0)	23.0 (21.0;26.0)	24.0 (21.0;26.0)	0.734	0.72	0.72
PTT, s	27.8 (24.9;33.6)	27.8 (24.9;33.6)	29.1 (25.8;36.2)	27.4 (24.7;32.2)	0.003	0.02	0.092
PT, s	13.4 (12.7;14.7)	13.4 (12.7;14.6)	13.7 (12.9;16.0)	13.3 (12.6;14.5)	0.002	0.001	0.543
INR	1.20 (1.10;1.30)	1.20 (1.10;1.30)	1.20 (1.10;1.50)	1.20 (1.10;1.30)	0.007	0.006	0.45
Weight, kg	75.0 (63.5;88.7)	75.0 (63.5;88.1)	70.0 (60.9;81.6)	76.7 (64.0;91.3)	<0.001	0.001	0.048
SOFA score	3.00 (1.00;4.00)	3.00 (1.00;4.00)	3.00 (2.00;5.00)	2.00 (1.00;4.00)	<0.001	<0.001	<0.001
SAPS score	33.0 (25.0;41.0)	33.0 (25.0;41.0)	37.0 (30.0;49.0)	30.0 (23.0;38.0)	<0.001	<0.001	<0.001
RRT	30 (1.04%)	17 (0.95%)	7 (2.42%)	6 (0.75%)	0.068	0.075	0.784
Ventilation	1239 (42.9%)	772 (43.0%)	119 (41.2%)	348 (43.4%)	0.803	0.897	0.898
Elixhauser score	5.00 (0.00;10.0)	5.00 (0.00;10.0)	5.00 (0.00;11.0)	5.00 (0.00;9.00)	0.77	0.763	0.87
CHD	414 (14.4%)	246 (13.7%)	56 (19.4%)	112 (14.0%)	0.036	0.043	0.912
Cardiac arrhythmias	960 (33.3%)	572 (31.9%)	103 (35.6%)	285 (35.5%)	0.126	0.346	0.224
Hypertension	1983 (68.7%)	1195 (66.6%)	193 (66.8%)	595 (74.2%)	<0.001	1	<0.001
Diabetes	672 (23.3%)	420 (23.4%)	92 (31.8%)	160 (20.0%)	<0.001	0.004	0.057
Liver disease	93 (3.22%)	56 (3.12%)	14 (4.84%)	23 (2.87%)	0.244	0.274	0.823
Peptic ulcer	14 (0.49%)	6 (0.33%)	7 (2.42%)	1 (0.12%)	<0.001	0.001	0.448
Solid tumor	47 (1.63%)	28 (1.56%)	8 (2.77%)	11 (1.37%)	0.261	0.294	0.848
Coagulopathy	139 (4.82%)	85 (4.74%)	15 (5.19%)	39 (4.86%)	0.944	0.97	0.97
Obesity	73 (2.53%)	38 (2.12%)	4 (1.38%)	31 (3.87%)	0.014	0.55	0.046
Length of hospital, days	6.78 (3.58;12.9)	6.91 (3.69;13.0)	4.91 (2.02;9.38)	7.18 (3.86;13.7)	<0.001	<0.001	0.157
Length of ICU stay, days	2.34 (1.22;5.84)	2.34 (1.21;6.08)	1.89 (1.05;3.72)	2.69 (1.44;6.00)	<0.001	<0.001	0.146
all-cause mortality	640 (22.2%)	370 (20.6%)	86 (29.8%)	184 (22.9%)	0.002	0.002	0.2
7-day mortality	490 (17.0%)	278 (15.5%)	74 (25.6%)	138 (17.2%)	<0.001	<0.001	0.298
28-day mortality	632 (21.9%)	366 (20.4%)	86 (29.8%)	180 (22.4%)	0.002	0.001	0.259

Data were expressed as *n* (%), mean (SD), or median (IQR). MAP, mean arterial pressure; GCS, Glasgow Coma Scale; WBC, white blood cell count; BUN, blood urea nitrogen; PTT, partial thromboplastin time; PT, prothrombin time; INR, international normalized ratio; SOFA, sequential organ failure assessment; SAPS, simplified acute physiology score; RRT, renal replacement therapy; CHD, chronic heart disease; bpm, beats per minute.

**Table 2 jcm-12-01556-t002:** Association between MAP and outcomes of stroke patients adjusted by multivariable Cox proportional hazards models.

Model		7-Day Mortality		28-Day Mortality	
		Hazard Ratio (95% CI)	*p*	Hazard Ratio (95% CI)	*p*
Model1	Low MAP	1.71(1.32–2.12)	<0.001	1.51(1.19–1.91)	<0.001
	High MAP	1.18(0.96–1.45)	0.120	1.18(0.99–1.41)	0.069
Model2	Low MAP	1.79(1.38–2.32)	<0.001	1.58(1.25–2.00)	<0.001
	High MAP	1.13(0.92–1.39)	0.239	1.12(0.93–1.34)	0.221
Model3	Low MAP	1.79(1.38–2.33)	<0.001	1.56(1.23–1.99)	<0.001
	High MAP	1.13(0.92–1.39)	0.241	1.12(0.94–1.34)	0.215
Model4	Low MAP	2.04(1.55–2.69)	<0.001	1.76(1.37–2.27)	<0.001
	High MAP	1.04(0.84–1.29)	0.702	1.09(0.90–1.32)	0.372
Model5	Low MAP	1.64(1.23–2.17)	0.001	1.51(1.17–1.94)	0.002
	High MAP	1.26(1.01–1.57)	0.043	1.28(1.06–1.55)	0.012

## Data Availability

This data used in this study are available at the database: https://mimic.mit.edu/.

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
