# Peer review of "Association between Mean Arterial Pressure during the First 24 Hours and Clinical Outcome in Critically Ill Stroke Patients: An Analysis of the MIMIC-III Database"

_jcm, 2023, doi:10.3390/jcm12041556_

Round 1

Reviewer 1 Report

This article is about the clinical outcomes of stroke patients (mainly ischemic stroke patients) that were divided into 3 different subcategories depending on the mean arterial pressure measured within first 24 hours. The of blood pressure management is extremely important for stroke patients and many works, including extensive meta-analysis, were devoted to it. However, the strategies and approaches for antihypertensive therapy vary depending on stroke type and time after stroke onset.

Line 63-65: «Although appropriate BP management has generally been demonstrated to be beneficial for stroke patients» - there are different guidelines for treatment of patients with ischemic and hemorrhage stroke, as well as for management of BP in acute and non-acute phase. I advise you to describe it more clearly.

Did you study mean arterial pressure during the first 24 hours, was antihypertensive therapy taken into account in any way during this period?

Line 240-244: «Guidelines suggest that intensive blood pressure reduction in acute stroke should be postponed for days or even weeks unless BP is grossly elevated (>220/120 mm Hg), or >200/100 with concomitant evidence of acute kidney injury, aortic dissection, cardiac ischemia, hypertensive encephalopathy or pulmonary oedema» There are different guidelines for ischemic and hemorrhage stroke (for example, 10.1161/STR.0000000000000407 and 10.1161/STR.0000000000000211) and recommendations are also different. In case of ischemic stroke, acute reduction of blood pressure is recommended in the case of mechanical thrombectomy, IV alteplase and for in-hospital management for «comorbid conditions (eg, concomitant acute coronary event, acute heart failure, aortic dissection, postfibrinolysis sICH, or preeclampsia/eclampsia)», «In patients with BP ≥220/120 mm Hg who did not receive IV alteplase or mechanical thrombectomy and have no comorbid conditions requiring urgent antihypertensive treatment, the benefit of initiating or reinitiating treatment of hypertension within the first 48 to 72 hours is uncertain. It might be reasonable to lower BP by 15% during the first 24 hours after onset of stroke». For acute hemorrhage: «In patients with spontaneous ICH requiring acute BP lowering, careful titration to ensure continuous smooth and sustained control of BP, avoiding peaks and large variability in SBP, can be beneficial for improving functional outcomes». In this regard, it is also necessary to describe the existing recommendations in more detail.

You write «stroke patients» and include only patients with «ICD-9 code ranged from 434 to 439».  The codes are written below:

434 Occlusion of cerebral arteries

435 Transient cerebral ischemia

436 Acute but ill-defined cerebrovascular disease

437 Other and ill-defined cerebrovascular disease

438 Late effects of cerebrovascular disease

439 ?

There is no code 430 Subarachnoid hemorrhage, 431 Intracerebral hemorrhage, 432 Other and unspecified intracranial hemorrhage and 433 Occlusion and stenosis of precerebral arteries (important: 433.0 Occlusion and stenosis of basilar artery). Please clarify these points.  Please explain what do you mean for patients with ICD-9 436 – ischemic stroke or hemorrhage? Or 438?

When you tested the models that you created, did you try to split the sample into 90-10% to verify the model on the remaining 10%?

Reviewer 2 Report

Although appropriate blood pressure (BP) management has generally been demonstrated to be beneficial for stroke patients, the optimal timing to initiate BP control and the targeted BP level in acute stroke remains to be elucidated. The authors investigated which MAP group showed lower mortality rate, low MAP, normal MAP, or high MAP group, using MIMIC-III database. As a result, 7-day and 28-day mortality was significantly higher in the low MAP group than that in the normal MAP group. In critically ill stroke patients, a low MAP significantly increased the 7-day and 28-day mortality, while a high MAP did not, suggesting that a low MAP is more harmful than a high MAP in critically ill stroke patients.

The manuscript is well written, several minor revisions should be required.

#1. Basically, sex is one of the fundamental factors which can make a difference in occurrence of stroke. Model 1 should include sex as well as age and ethnicity in Cox regression analysis.

#2. In Table 1, RRT, hypertension, diabetes, solid tumor, coagulopathy, and obesity are included as confounders. The authors should indicate each definition.

#3. In Table 1, two ‘p value for normal’ are shown next to ‘p value overall’. I guess that two ‘p value for normal’ are some mistake. Correct these words.

#4. Show p for non-linearity in Figure 3. If they are not statistically significant, the results were not consistent.

#5. The authors explain that the association between MAP and mortality was consistent in the following 201 subgroups: age > 60 or ⩽60, male or female, with or without hypertension, with or without 202 diabetes, and with or without CHD in lines 201-203. However, some are statistically significant, some are not. Therefore, the associations are not consistent. The authors should use appropriate wording.

#6. The authors indicate that they re-performed restricted cubic spline analyses in the same above-mentioned subgroups, the approximately L-shaped association between MAP and the 7-day and 28-day mortality remained robust across different subgroups (Figure 5, Figure 6), in lines 203-206. Show p for non-linearity in each graph. If some do not show significance, the approximately L-shaped associations are not robust. The authors should use appropriate wording.
